# Conformalized Quantile Regression

**Yaniv Romano**
Department of Statistics
Stanford University

**Evan Patterson**
Department of Statistics
Stanford University

**Emmanuel J. Candès**
Departments of Mathematics and of Statistics
Stanford University

## Abstract

Conformal prediction is a technique for constructing prediction intervals that attain valid coverage in finite samples, without making distributional assumptions. Despite this appeal, existing conformal methods can be unnecessarily conservative because they form intervals of constant or weakly varying length across the input space. In this paper we propose a new method that is fully adaptive to heteroscedasticity. It combines conformal prediction with classical quantile regression, inheriting the advantages of both. We establish a theoretical guarantee of valid coverage, supplemented by extensive experiments on popular regression datasets. We compare the efficiency of conformalized quantile regression to other conformal methods, showing that our method tends to produce shorter intervals.

## 1   Introduction

In many applications of regression modeling, it is important not only to predict accurately but also to quantify the accuracy of the predictions. This is especially true in situations involving high-stakes decision making, such as estimating the efficacy of a drug or the risk of a credit default. The uncertainty in a prediction can be quantified using a prediction interval, giving lower and upper bounds between which the response variable lies with high probability. An ideal procedure for generating prediction intervals should satisfy two properties. First, it should provide valid coverage in finite samples, without making strong distributional assumptions, such as Gaussianity. Second, its intervals should be as short as possible at each point in the input space, so that the predictions will be informative. When the data is heteroscedastic, getting valid but short prediction intervals requires adjusting the lengths of the intervals according to the local variability at each query point in predictor space. This paper introduces a procedure that performs well on both criteria, being distribution-free and adaptive to heteroscedasticity.

Our work is heavily inspired by *conformal prediction*, a general methodology for constructing prediction intervals [1–6]. Conformal prediction has the virtue of providing a nonasymptotic, distribution-free coverage guarantee. The main idea is to fit a regression model on the training samples, then use the residuals on a held-out validation set to quantify the uncertainty in future predictions. The effect of the underlying model on the length of the prediction intervals, and attempts to construct intervals with locally varying length, have been studied in numerous recent works [6–16]. Nevertheless, existing methods yield conformal intervals of either fixed length or length depending only weakly on the predictors, as argued in [6, 15, 17].

In conformal prediction to date, there has been a mismatch between the primary inferential focus—conditional mean estimation—and the ultimate inferential goal—prediction interval estimation. Statistical efficiency is lost by estimating a mean when an interval is needed. A more direct approach

to interval estimation is offered by quantile regression [18]. Take any algorithm for *quantile regression*, i.e., for estimating conditional quantile functions from data. To obtain prediction intervals with, say, nominal 90% coverage, simply fit the conditional quantile function at the 5% and 95% levels and form the corresponding intervals. Even for highly heteroscedastic data, this methodology has been shown to be adaptive to local variability [19–25]. However, the validity of the estimated intervals is guaranteed only for specific models, under certain regularity and asymptotic conditions [22–24].

In this work, we combine conformal prediction with quantile regression. The resulting method, which we call *conformalized quantile regression* (CQR), inherits both the finite sample, distribution-free validity of conformal prediction and the statistical efficiency of quantile regression.[1] On one hand, CQR is flexible in that it can wrap around any algorithm for quantile regression, including random forests and deep neural networks [26–29]. On the other hand, a key strength of CQR is its rigorous control of the miscoverage rate, independent of the underlying regression algorithm.

**Summary and outline**

Suppose we are given $n$ training samples $\{(X_i, Y_i)\}_{i=1}^n$ and we must now predict the unknown value of $Y_{n+1}$ at a test point $X_{n+1}$. We assume that all the samples $\{(X_i, Y_i)\}_{i=1}^{n+1}$ are drawn exchangeably—for instance, they may be drawn i.i.d.—from an arbitrary joint distribution $P_{XY}$ over the feature vectors $X \in \mathbb{R}^p$ and response variables $Y \in \mathbb{R}$. We aim to construct a *marginal distribution-free prediction interval* $C(X_{n+1}) \subseteq \mathbb{R}$ that is likely to contain the unknown response $Y_{n+1}$. That is, given a desired miscoverage rate $\alpha$, we ask that

$$\mathbb{P}\{Y_{n+1} \in C(X_{n+1})\} \geq 1 - \alpha \tag{1}$$

for any joint distribution $P_{XY}$ and any sample size $n$. The probability in this statement is marginal, being taken over all the samples $\{(X_i, Y_i)\}_{i=1}^{n+1}$.

To accomplish this, we build on the method of conformal prediction [2, 3, 8]. We first split the training data into two disjoint subsets, a proper training set and a calibration set.[2] We fit two quantile regressors on the proper training set to obtain initial estimates of the lower and upper bounds of the prediction interval, as explained in Section 2. Then, using the calibration set, we conformalize and, if necessary, correct this prediction interval. Unlike the original interval, the conformalized prediction interval is guaranteed to satisfy the coverage requirement (1) regardless of the choice or accuracy of the quantile regression estimator. We prove this in Section 4.

Our method differs from the standard method of conformal prediction [3, 15], recalled in Section 3, in that we calibrate the prediction interval using conditional quantile regression, while the standard method uses only classical, conditional mean regression. The result is that our intervals are adaptive to heteroscedasticity whereas the standard intervals are not. We evaluate the statistical efficiency of our framework by comparing its miscoverage rate and average interval length with those of other methods. We review existing state-of-the-art schemes for conformal prediction in Section 5 and we compare them with our method in Section 6. Based on extensive experiments across eleven datasets, we conclude that conformal quantile regression yields shorter intervals than the competing methods.

## 2   Quantile regression

The aim of conditional quantile regression [18] is to estimate a given quantile, such as the median, of $Y$ conditional on $X$. Recall that the *conditional distribution function* of $Y$ given $X = x$ is

$$F(y \mid X = x) := \mathbb{P}\{Y \leq y \mid X = x\},$$

and that the $\alpha$th *conditional quantile function* is

$$q_\alpha(x) := \inf\{y \in \mathbb{R} : F(y \mid X = x) \geq \alpha\}.$$

Fix the lower and upper quantiles to be equal to $\alpha_{\mathrm{lo}} = \alpha/2$ and $\alpha_{\mathrm{hi}} = 1 - \alpha/2$, say. Given the pair $q_{\alpha_{\mathrm{lo}}}(x)$ and $q_{\alpha_{\mathrm{hi}}}(x)$ of lower and upper conditional quantile functions, we obtain a conditional prediction interval for $Y$ given $X = x$, with miscoverage rate $\alpha$, as

$$C(x) = [q_{\alpha_{\mathrm{lo}}}(x),\ q_{\alpha_{\mathrm{hi}}}(x)]. \tag{2}$$

By construction, this interval satisfies

$$\mathbb{P}\{Y \in C(X) | X = x\} \geq 1 - \alpha. \tag{3}$$

Notice that the length of the interval $C(X)$ can vary greatly depending on the value of $X$. The uncertainty in the prediction of $Y$ is naturally reflected in the length of the interval. In practice we cannot know this ideal prediction interval, but we can try to estimate it from the data.

**Estimating quantiles from data**

Classical regression analysis estimates the conditional mean of the test response $Y_{n+1}$ given the features $X_{n+1}=x$ by minimizing the sum of squared residuals on the $n$ training points:

$$\hat{\mu}(x) = \mu(x; \hat{\theta}), \qquad \hat{\theta} = \underset{\theta}{\operatorname{argmin}} \frac{1}{n} \sum_{i=1}^{n} (Y_i - \mu(X_i; \theta))^2 + \mathcal{R}(\theta).$$

Here $\theta$ are the parameters of the regression model, $\mu(x; \theta)$ is the regression function, and $\mathcal{R}$ is a potential regularizer.

Analogously, quantile regression estimates a conditional quantile function $q_\alpha$ of $Y_{n+1}$ given $X_{n+1}=x$. This can be cast as the optimization problem

$$\hat{q}_\alpha(x) = f(x; \hat{\theta}), \qquad \hat{\theta} = \underset{\theta}{\operatorname{argmin}} \frac{1}{n} \sum_{i=1}^{n} \rho_\alpha(Y_i, f(X_i; \theta)) + \mathcal{R}(\theta),$$

where $f(x; \theta)$ is the quantile regression function and the loss function $\rho_\alpha$ is the "check function" or "pinball loss" [18, 24], defined by

$$\rho_\alpha(y, \hat{y}) := \begin{cases} \alpha(y - \hat{y}) & \text{if } y - \hat{y} > 0, \\ (1 - \alpha)(\hat{y} - y) & \text{otherwise.} \end{cases}$$

The simplicity and generality of this formulation makes quantile regression widely applicable. As in classical regression, one can leverage the great variety of machine learning methods to design and learn $\hat{q}_\alpha$ [19–21, 23, 30].

All this suggests an obvious strategy to construct a prediction band with nominal miscoverage rate $\alpha$: estimate $\hat{q}_{\alpha_{\mathrm{lo}}}(x)$ and $\hat{q}_{\alpha_{\mathrm{hi}}}(x)$ using quantile regression, then output $\hat{C}(X_{n+1}) = [\hat{q}_{\alpha_{\mathrm{lo}}}(X_{n+1}), \hat{q}_{\alpha_{\mathrm{hi}}}(X_{n+1})]$ as an estimate of the ideal interval $C(X_{n+1})$ from equation (2). This approach is widely applicable and often works well in practice, yielding intervals that are adaptive to heteroscedasticity. However, it is not guaranteed to satisfy the coverage statement (3) when $C(X)$ is replaced by the estimated interval $\hat{C}(X_{n+1})$. Indeed, the absence of any finite sample guarantee can sometimes be disastrous. This worry is corroborated by our experiments, which show that the intervals constructed by neural networks can substantially undercover.

Under regularity conditions and for specific models, estimates of conditional quantile functions via the pinball loss or related methods are known to be asymptotically consistent [23, 24, 31, 32]. Certain methods that do not minimize the pinball loss, such as quantile random forests [22], are also asymptotically consistent. But to get valid coverage in finite samples, we must draw on a different set of ideas, from conformal prediction.

## 3   Conformal Prediction

We now describe how conformal prediction [1, 3] constructs prediction intervals that satisfy the finite-sample coverage guarantee (1). To be carried out exactly, the original, or *full*, conformal procedure effectively requires the regression algorithm to be invoked infinitely many times. In contrast, the method of *split*, or *inductive*, conformal prediction [2, 8] avoids this problem, at the cost of splitting the data. While our proposal is applicable to both versions of conformal prediction, in the interest of space we will restrict our attention to split conformal prediction and refer the reader to [3, 15] for a more detailed comparison between the two methods.

Under the assumptions of Section 1, the split conformal method begins by splitting the training data into two disjoint subsets: a proper training set $\{(X_i, Y_i) : i \in \mathcal{I}_1\}$ and calibration set

$\{(X_i, Y_i) : i \in \mathcal{I}_2\}$. Then, given any regression algorithm $\mathcal{A}$,[3] a regression model is fit to the proper training set:

$$\hat{\mu}(x) \leftarrow \mathcal{A}\left(\{(X_i, Y_i) : i \in \mathcal{I}_1\}\right).$$

Next, the absolute residuals are computed on the calibration set, as follows:

$$R_i = |Y_i - \hat{\mu}(X_i)|, \qquad i \in \mathcal{I}_2. \tag{4}$$

For a given level $\alpha$, we then compute a quantile of the empirical distribution[4] of the absolute residuals,

$$Q_{1-\alpha}(R, \mathcal{I}_2) := (1-\alpha)(1 + 1/|\mathcal{I}_2|)\text{-th empirical quantile of } \{R_i : i \in \mathcal{I}_2\}.$$

Finally, the prediction interval at a new point $X_{n+1}$ is given by

$$C(X_{n+1}) = [\hat{\mu}(X_{n+1}) - Q_{1-\alpha}(R, \mathcal{I}_2), \ \hat{\mu}(X_{n+1}) + Q_{1-\alpha}(R, \mathcal{I}_2)]. \tag{5}$$

This interval is guaranteed to satisfy (1), as shown in [3]. For related theoretical studies, see [15, 33].

A closer look at the prediction interval (5) reveals a major limitation of this procedure: the length of $C(X_{n+1})$ is fixed and equal to $2Q_{1-\alpha}(R, \mathcal{I}_2)$, independent of $X_{n+1}$. Lei et al [15] observe that the intervals produced by the full conformal method also vary only slightly with $X_{n+1}$, provided the regression algorithm is moderately stable. This brings us to our proposal, which offers a principled approach to constructing variable-width conformal prediction intervals.

## 4  Conformalized quantile regression (CQR)

In this section we introduce our procedure, beginning with a small experiment on simulated data to show how it improves upon standard conformal prediction. Figure 1 compares the prediction intervals produced by (a) the split conformal method, (b) its locally adaptive variant (described later in Section 5), and (c) our method, conformalized quantile regression (CQR). The heteroskedasticity of the data is evident, as the dispersion of $Y$ varies considerably with $X$. The data also contains outliers, shown in the supplementary material. For all three methods, we construct $90\%$ prediction intervals on the test data. From Figures 1a and 1d, we see that the lengths of the split conformal intervals are fixed and equal to 2.91. The prediction intervals of the locally weighted variant, shown in Figure 1b, are partially adaptive, resulting in slightly shorter intervals, of average length 2.86. Our method, shown in Figure 1c, is also adaptive, but its prediction intervals are considerably shorter, of average length 1.99, due to better estimation of the lower and upper quantiles. We refer the reader to the supplementary material for further details about this experiment, as well as a second simulation demonstrating the advantage of CQR on heavy-tailed data.

We now describe CQR itself. As in split conformal prediction, we begin by splitting the data into a proper training set, indexed by $\mathcal{I}_1$, and a calibration set, indexed by $\mathcal{I}_2$. Given any quantile regression algorithm $\mathcal{A}$, we then fit two conditional quantile functions $\hat{q}_{\alpha_{\mathrm{lo}}}$ and $\hat{q}_{\alpha_{\mathrm{hi}}}$ on the proper training set:

$$\{\hat{q}_{\alpha_{\mathrm{lo}}}, \hat{q}_{\alpha_{\mathrm{hi}}}\} \leftarrow \mathcal{A}(\{(X_i, Y_i) : i \in \mathcal{I}_1\}).$$

In the essential next step, we compute *conformity scores* that quantify the error made by the plug-in prediction interval $\hat{C}(x) = [\hat{q}_{\alpha_{\mathrm{lo}}}(x), \ \hat{q}_{\alpha_{\mathrm{hi}}}(x)]$. The scores are evaluated on the calibration set as

$$E_i := \max\{\hat{q}_{\alpha_{\mathrm{lo}}}(X_i) - Y_i, Y_i - \hat{q}_{\alpha_{\mathrm{hi}}}(X_i)\} \tag{6}$$

for each $i \in \mathcal{I}_2$. The conformity score $E_i$ has the following interpretation. If $Y_i$ is below the lower endpoint of the interval, $Y_i < \hat{q}_{\alpha_{\mathrm{lo}}}(X_i)$, then $E_i = |Y_i - \hat{q}_{\alpha_{\mathrm{lo}}}(X_i)|$ is the magnitude of the error incurred by this mistake. Similarly, if $Y_i$ is above the upper endpoint of the interval, $Y_i > \hat{q}_{\alpha_{\mathrm{hi}}}(X_i)$, then $E_i = |Y_i - \hat{q}_{\alpha_{\mathrm{hi}}}(X_i)|$. Finally, if $Y_i$ correctly belongs to the interval, $\hat{q}_{\alpha_{\mathrm{lo}}}(X_i) \le Y_i \le \hat{q}_{\alpha_{\mathrm{hi}}}(X_i)$, then $E_i$ is the larger of the two non-positive numbers $\hat{q}_{\alpha_{\mathrm{lo}}}(X_i) - Y_i$ and $Y_i - \hat{q}_{\alpha_{\mathrm{hi}}}(X_i)$ and so is itself non-positive. The conformity score thus accounts for both undercoverage and overcoverage.

Finally, given new input data $X_{n+1}$, we construct the prediction interval for $Y_{n+1}$ as

$$C(X_{n+1}) = [\hat{q}_{\alpha_{\mathrm{lo}}}(X_{n+1}) - Q_{1-\alpha}(E, \mathcal{I}_2), \ \hat{q}_{\alpha_{\mathrm{hi}}}(X_{n+1}) + Q_{1-\alpha}(E, \mathcal{I}_2)], \tag{7}$$

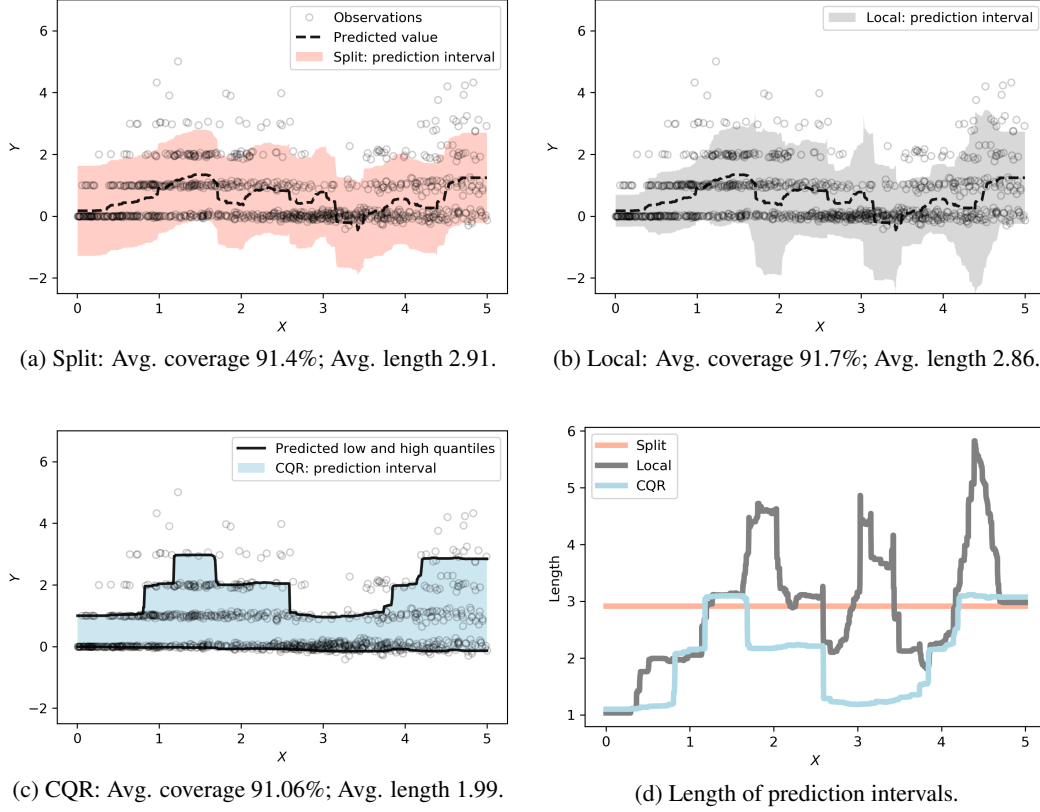

(a) Split: Avg. coverage 91.4%; Avg. length 2.91.

(b) Local: Avg. coverage 91.7%; Avg. length 2.86.

(c) CQR: Avg. coverage 91.06%; Avg. length 1.99.

(d) Length of prediction intervals.

Figure 1: Prediction intervals on simulated heteroscedastic data with outliers (see the supplementary material for a full range display): (a) the standard split conformal method, (b) its locally adaptive variant, and (c) CQR (our method). The length of the interval as a function of $X$ is shown in (d). The target coverage rate is 90%. The broken black curve in (a) and (b) is the pointwise prediction from the random forest estimator. In (c), we show two curves, representing the lower and upper quantile regression estimates based on random forests [22]. Observe how in this example the quantile regression estimates closely match the adjusted estimates—the boundary of the blue region—obtained by conformalization.

---

**Algorithm 1:** Split Conformal Quantile Regression.

---

**Input:**
  Data $(X_i, Y_i)$, $1 \leq i \leq n$; miscoverage level $\alpha \in (0, 1)$; quantile regression algorithm $\mathcal{A}$.
**Process:**
  Randomly split $\{1, \ldots, n\}$ into two disjoint sets $\mathcal{I}_1$ and $\mathcal{I}_2$.
  Fit two conditional quantile functions: $\{\hat{q}_{\alpha_{\mathrm{lo}}}, \hat{q}_{\alpha_{\mathrm{hi}}}\} \leftarrow \mathcal{A}(\{(X_i, Y_i) : i \in \mathcal{I}_1\})$.
  Compute $E_i$ for each $i \in \mathcal{I}_2$, as in equation (6).
  Compute $Q_{1-\alpha}(E, \mathcal{I}_2)$, the $(1-\alpha)(1 + 1/|\mathcal{I}_2|)$-th empirical quantile of $\{E_i : i \in \mathcal{I}_2\}$.
**Output:**
  Prediction interval $C(x) = [\hat{q}_{\alpha_{\mathrm{lo}}}(x) - Q_{1-\alpha}(E, \mathcal{I}_2), \ \hat{q}_{\alpha_{\mathrm{hi}}}(x) + Q_{1-\alpha}(E, \mathcal{I}_2)]$ for $X_{n+1} = x$.

---

where

$$Q_{1-\alpha}(E, \mathcal{I}_2) := (1-\alpha)(1 + 1/|\mathcal{I}_2|)\text{-th empirical quantile of } \{E_i : i \in \mathcal{I}_2\} \quad (8)$$

conformalizes the plug-in prediction interval.

For ease of reference, the CQR procedure is summarized in Algorithm 1. The following theorem, establishing its validity, is proved in the supplementary material.

**Theorem 1.** *If $(X_i, Y_i)$, $i = 1, \ldots, n+1$ are exchangeable, then the prediction interval $C(X_{n+1})$ constructed by the split CQR algorithm satisfies*

$$\mathbb{P}\{Y_{n+1} \in C(X_{n+1})\} \geq 1 - \alpha.$$

*Moreover, if the conformity scores $E_i$ are almost surely distinct, then the prediction interval is nearly perfectly calibrated:*

$$\mathbb{P}\{Y_{n+1} \in C(X_{n+1})\} \leq 1 - \alpha + 1/(|\mathcal{I}_2| + 1).$$

**Practical considerations and extensions**

Conformalized quantile regression can accommodate a wide range of quantile regression methods [18–23, 25, 30] to estimate the conditional quantile functions, $q_{\alpha_{lo}}$ and $q_{\alpha_{hi}}$. The estimators can be even be aggregates of different quantile regression algorithms. Recently, new deep learning techniques have been proposed [26–29] for constructing prediction intervals. These methods could be wrapped by our framework and would then immediately enjoy rigorous coverage guarantees. In our experiments, we focus on quantile neural networks [20] and quantile regression forests [22].

Because the underlying quantile regression algorithm may process the proper training set in arbitrary ways, our framework affords broad flexibility in hyper-parameter tuning. Consider, for instance, the tuning of typical hyper-parameters of neural networks, such as the batch size, the learning rate, and the number of epochs. The hyperparameters may be selected, as usual, by cross validation, where we minimize the average interval length over the folds.

In this vein, we record two specific implementation details that we have found to be useful.

1. Quantile regression is sometimes *too* conservative, resulting in unnecessarily wide prediction intervals. In our experience, quantile regression forests [22] are often overly conservative and quantile neural networks [20] are occasionally so. We can mitigate this problem by tuning the nominal quantiles of the underlying method as additional hyper-parameters in cross validation. Notably, this tuning does not invalidate the coverage guarantee, but it may yield shorter intervals, as our experiments confirm.

2. To reduce the computational cost, instead of fitting two separate neural networks to estimate the lower and upper quantile functions, we can replace the standard one-dimensional estimate of the unknown response by a two-dimensional estimate of the lower and upper quantiles. In this way, most of the network parameters are shared between the two quantile estimators. We adopt this approach in the experiments of Section 6.

Another avenue for extension is the conformalization step. The conformalization implemented by equations (7) and (8) allows coverage errors to be spread arbitrarily over the left and right tails. Using a method reminiscent of [34], we can control the left and right tails independently, yielding a stronger coverage guarantee. It is stated below and proved in the supplementary material. As we will see in Section 6, the price paid for the stronger coverage guarantee is slightly longer intervals.

**Theorem 2.** *Define the prediction interval*

$$C(X_{n+1}) := [\hat{q}_{\alpha_{lo}}(X_{n+1}) - Q_{1-\alpha_{lo}}(E_{lo}, \mathcal{I}_2), \ \hat{q}_{\alpha_{hi}}(X_{n+1}) + Q_{1-\alpha_{hi}}(E_{hi}, \mathcal{I}_2)],$$

*where $Q_{1-\alpha_{lo}}(E_{lo}, \mathcal{I}_2)$ is the $(1 - \alpha_{lo})$-th empirical quantile of $\{\hat{q}_{\alpha_{lo}}(X_i) - Y_i : i \in \mathcal{I}_2\}$ and $Q_{1-\alpha_{hi}}(E_{hi}, \mathcal{I}_2)$ is the $(1 - \alpha_{hi})$-th empirical quantile of $\{Y_i - \hat{q}_{\alpha_{hi}}(X_i) : i \in \mathcal{I}_2\}$. If the samples $(X_i, Y_i)$, $i = 1, \ldots, n+1$ are exchangeable, then*

$$\mathbb{P}\{Y_{n+1} \geq \hat{q}_{\alpha_{lo}}(X_{n+1}) - Q_{1-\alpha_{lo}}(E_{lo}, \mathcal{I}_2)\} \geq 1 - \alpha_{lo}$$

*and*

$$\mathbb{P}\{Y_{n+1} \leq \hat{q}_{\alpha_{hi}}(X_{n+1}) + Q_{1-\alpha_{hi}}(E_{hi}, \mathcal{I}_2)\} \geq 1 - \alpha_{hi}.$$

*Consequently, assuming $\alpha = \alpha_{lo} + \alpha_{hi}$, we also have $\mathbb{P}\{Y_{n+1} \in C(X_{n+1})\} \geq 1 - \alpha$.*

## 5 Related work: locally adaptive conformal prediction

*Locally adaptive split conformal prediction*, first proposed in [7, 9] and later studied in [15], is an earlier approach to making conformal prediction adaptive to heteroskedascity. Like our method, it

starts from the observation that one can replace the absolute residuals in equation (4) by any other loss function that treats the data exchangeably. In this case, the absolute residuals $R_i$ are replaced by the scaled residuals $\tilde{R}_i := |Y_i - \hat{\mu}(X_i)|/\hat{\sigma}(X_i) = R_i/\hat{\sigma}(X_i), \ i \in \mathcal{I}_2$, where $\hat{\sigma}(X_i)$ is a measure of the dispersion of the residuals at $X_i$. Usually $\hat{\sigma}(x)$ is an estimate of the conditional mean absolute deviation (MAD) of $|Y - \hat{\mu}(x)|$ given $X = x$. Finally, the prediction interval at $X_{n+1}$ is computed as $C(X_{n+1}) = \left[ \hat{\mu}(X_{n+1}) - \hat{\sigma}(X_{n+1})Q_{1-\alpha}(\tilde{R}, \mathcal{I}_2), \ \hat{\mu}(X_{n+1}) + \hat{\sigma}(X_{n+1})Q_{1-\alpha}(\tilde{R}, \mathcal{I}_2) \right]$. Both $\hat{\mu}$ and $\hat{\sigma}$ are fit only on the proper training set. Consequently, $\hat{\mu}$ and $\hat{\sigma}$ satisfy the assumptions of conformal prediction and, hence, locally adaptive conformal prediction inherits the coverage guarantee of standard conformal prediction.

In practice, locally adaptive conformal prediction requires fitting two functions, in sequence, on the proper training set. (Thus it is more computationally expensive than standard conformal prediction.) First, one fits the conditional mean function $\hat{\mu}(x)$, as described in Section 3. Then one fits $\hat{\sigma}(x)$ to the pairs $\{(X_i, R_i) : i \in \mathcal{I}_1\}$, using a regression model that predicts the residuals $R_i$ given the inputs $X_i$. As an example, the intervals in Figure 1b above are created by locally adaptive split conformal prediction, where both $\hat{\mu}$ and $\hat{\sigma}$ are random forests.

Locally adaptive conformal prediction is limited in several ways, some more important than others. A first limitation, already noted in [15], appears when the data is actually homoskedastic. In this case, the locally adaptive method suffers from inflated prediction intervals compared to the standard method. This is presumably due to the extra variability introduced by estimating $\hat{\sigma}$ as well as $\hat{\mu}$.

The locally adaptive method faces a more fundamental statistical limitation. There is an essential difference between the residuals on the proper training set and the residuals on the calibration set: the former are biased by an optimization procedure designed to minimize them, while the latter are unbiased. Because it uses the proper training residuals (as it must to ensure valid coverage), the locally adaptive method tends to systematically underestimate the prediction error. In general, this forces the correction constant $Q_{1-\alpha}(\tilde{R}, \mathcal{I}_2)$ to be large and the intervals to be less adaptive.

To press this point further, suppose the conditional mean function $\hat{\mu}$ is a deep neural network. It is well attested in the deep learning literature that, given enough training samples, the best prediction error is attained by "over-fitting" to the training data, in the sense that the training error is nearly zero. The training residuals are then very poor estimates of the true prediction error, resulting in severe loss of adaptivity. Our method, in contrast, does not suffer from this problem because the original training objective is to estimate the lower and upper conditional quantiles, not the conditional mean.

## 6 Experiments

In this section we systematically compare our method, conformalized quantile regression, to the standard and locally adaptive versions of split conformal prediction. Among preexisting conformal prediction algorithms, we select leading variants that use random forests [10] and neural networks [35] for conditional mean regression. Specifically, we evaluate the original version of split conformal prediction (Section 3) using three regression algorithms: **Ridge**, **Random Forests** and **Neural Net**. We evaluate locally adaptive conformal prediction (Section 5) using the same three underlying regression algorithms: **Ridge Local**, **Random Forests Local**, and **Neural Net Local**. Likewise, we configure our method (Algorithm 1) to use quantile random forests [22], **CQR Random Forests**, and quantile neural networks [20], **CQR Neural Net**. Finally, as a baseline, we also include the previous two quantile regression algorithms, but without any conformalization: **Quantile Random Forests** and **Quantile Neural Net**. The last two methods, in contrast to the others, do not have finite-sample coverage guarantees. All implementation details are available in the supplementary material.

We conduct the experiments on eleven benchmark datasets for regression, listed in the supplementary material. In each case, we standardize the features to have zero mean and unit variance and we rescale the response by dividing it by its mean absolute value.[5] The performance metrics are averaged over 20 different training-test splits; $80\%$ of the examples are used for training and the remaining $20\%$

| Method | Avg. Length | Avg. Coverage |
|---|---|---|
| Ridge | 3.07 | 90.08 |
| Ridge Local | 2.93 | 90.14 |
| Random Forests | 2.24 | 90.00 |
| Random Forests Local | 1.82 | 89.99 |
| Neural Net | 2.20 | 89.95 |
| Neural Net Local | 1.79 | 90.02 |
| **CQR Random Forests** | **1.40** | **90.34** |
| **CQR Neural Net** | **1.40** | **90.02** |
| *Quantile Random Forests | *2.21 | *92.62 |
| *Quantile Neural Net | *1.50 | *88.87 |

Table 1: Length and coverage of prediction intervals ($\alpha = 0.1$), averaged across 11 datasets and 20 random training-test splits. Our methods are shown in bold font. The methods marked by an asterisk are not supported by finite-sample coverage guarantees.

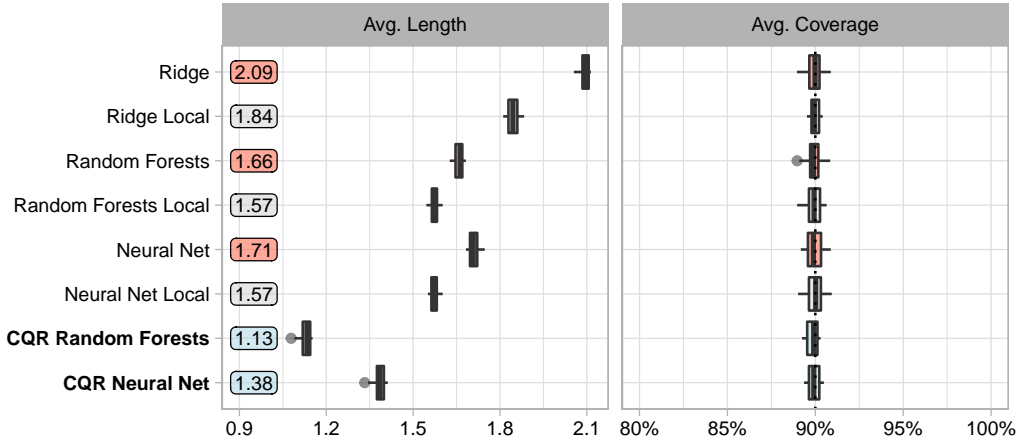

Figure 2: Average length (left) and coverage (right) of prediction intervals ($\alpha = 0.1$) on the `bio` dataset [36]. The numbers in the colored boxes are the average lengths, shown in red for split conformal, in gray for locally adapative split conformal, and in light blue for our method.

for testing. The proper training and calibration sets for split conformal prediction have equal size. Throughout the experiments the nominal miscoverage rate is fixed and set to $\alpha = 0.1$.

Table 1 summarizes our 2,200 experiments, showing the average performance across all the datasets and training-test splits. On average, our method achieves shorter prediction intervals than both standard and locally adaptive conformal prediction. It may seem surprising that our method also outperforms non-conformalized quantile regression, which is permitted more training data. There are several possible explanations for this. First, the non-conformalized methods sometimes *over*cover, but that is mitigated by our signed conformity scores (6). In addition, by using CQR, we can tune the quantiles of the underlying quantile regression algorithms using cross-validation (Section 4). Interestingly, CQR selects quantiles below the nominal level.

Turning to the issue of valid coverage, all methods based on conformal prediction successfully construct prediction bands at the nominal coverage rate of $90\%$, as the theory suggests they should. One of the non-conformalized methods, based on random forests, is slightly conservative, while the other, based on neural networks, tends to undercover. In fact, other authors have shown that the coverage of quantile neural networks depends greatly on the tuning of the hyper-parameters, with, for instance, the actual coverage in [25, Figure 3] ranging from the nominal $95\%$ to well below $50\%$. Such volatility demonstrates the importance of the conformal prediction's finite-sample guarantee.

When estimating a lower and an upper quantile by two separate quantile regressions, there is no guarantee that the lower estimate will actually be smaller than the upper estimate. This is known as

the *quantile crossing* problem [37]. Quantile crossing can affect quantile neural networks, but not quantile regression forests. When the two quantiles are far apart, as in the 5% and 95% quantiles, we should expect the estimates to cross very infrequently and that is indeed what we find in the experiments. Nevertheless, we also evaluated a post-processing method to eliminate crossings [38]. It yields a slight improvement in performance: the average interval length of the CQR neural networks drops from 1.40 to 1.35, while the coverage rate remains the same. The average interval length of the unconformalized quantile neural networks drops from 1.50 to 1.40, with a decrease in the average coverage rate, from 88.87 to 87.99.

As expected, adopting the two-tailed, asymmetric conformalization proposed in Theorem 2 causes an increase in average interval length compared to the symmetric conformalization of Theorem 1. Specifically, the average length for CQR neural networks increases from 1.40 to 1.58, while the coverage rate stays about the same. The average length for the CQR random forests increases from 1.40 to 1.57, accompanied by a slight increase in the average coverage rate, from 90.34 to 90.94.

In a series of figures, provided in the supplementary material, we break down the performance of the different methods on each of the benchmark datasets. The performance on individual datasets confirms the overall trend in Table 1. Locally adaptive conformal prediction generally outperforms standard conformal prediction, and, on ten out of eleven datasets, conformalized quantile regression outperforms both. As a representative example, Figure 2 shows our results on a dataset (`bio`) about the physicochemical properties of protein tertiary structure [36].

# 7    Conclusion

Conformal quantile regression is a new way of constructing prediction intervals that combines the advantages of conformal prediction and quantile regression. It provably controls the miscoverage rate in finite samples, under the mild distributional assumption of exchangeability, while adapting the interval lengths to heteroskedasticity in the data.

We expect the ideas behind conformal quantile regression to be applicable in the related setting of conformal predictive distributions [39]. In this extension of conformal prediction, the aim is to estimate a predictive probability distribution, not just an interval. We see intriguing connections between our work and a very recent, independently written paper on conformal distributions [17].

**Acknowledgements**

E. C. was partially supported by the Office of Naval Research (ONR) under grant N00014-16- 1-2712, by the Army Research Office (ARO) under grant W911NF-17-1-0304, by the Math + X award from the Simons Foundation and by a generous gift from TwoSigma. E. P. and Y. R. were partially supported by the ARO grant. Y. R. was also supported by the same Math + X award. Y. R. thanks the Zuckerman Institute, ISEF Foundation and the Viterbi Fellowship, Technion, for providing additional research support. We thank Chiara Sabatti for her insightful comments on a draft of this paper and Ryan Tibshirani for his crucial remarks on our early experimental findings.

## Footnotes

[1] An implementation of CQR is available online at `https://github.com/yromano/cqr`.

[2] Like conformal regression, CQR has a variant that does not require data splitting.

[3] In full conformal prediction, the regression algorithm must treat the data exchangeably, but no such restrictions apply to split conformal prediction.

[4] The explicit formula for empirical quantiles is recalled in the supplementary material.

[5]In the experiments, we compute the needed sample means and variances only on the proper training set. This ensures that if the original data is exchangeable, then the rescaled data remains so. That being said, we could also rescale using sample means and variances computed on the test data, because it would preserve exchangeability even while it destroys independence.

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
