[Supplementary Material]

# Supplementary Material for
# Conformalized Quantile Regression

**Yaniv Romano**
Department of Statistics
Stanford University

**Evan Patterson**
Department of Statistics
Stanford University

**Emmanuel J. Candès**
Departments of Mathematics and of Statistics
Stanford University

## A  Synthetic experiments

### A.1  Heteroscedastic data with outliers

Figure 1: Full range scatter plot of the test data used in the synthetic simulation of Section 4.

In Section 4, we presented an experiment on simulated data to illustrate the importance of adaptivity in conformal prediction. Here we describe the details of that experiment.

To generate the training data, we draw $n = 2000$ independent, univariate predictor samples $X_i$ from the uniform distribution on the interval $[1, 5]$. The response variable is then sampled as

$$Y_i \sim \text{Pois}(\sin^2(X_i) + 0.1) + 0.03\, X_i\, \epsilon_{1,i} + 25\; \mathbb{1}\{U_i < 0.01\}\, \epsilon_{2,i}, \qquad (1)$$

where $\text{Pois}(\lambda)$ is the Poisson distribution with mean $\lambda$, both $\epsilon_{1,i}$ and $\epsilon_{2,i}$ are i.i.d. standard Gaussian noise, and the $U_i$ are uniform on the interval $[0, 1]$. We generate a test set of 5000 samples in the same way. The last term in equation (1) creates few but large outliers. This is illustrated in Figure 1, which plots the synthetic data across its full range.

In the synthetic experiment of Section 4, we construct a $90\%$ prediction interval for the test data using split conformal prediction. Specifically, we split the training data into two subsets, train a random forest regressor on the first set, and calibrate the intervals on the second set. We do the same for locally adaptive split conformal prediction. The scale estimator is another random forest. The experiment is insensitive to the value of the hyper-parameter $\gamma$; we set it to zero. Finally, we instantiate our method, conformal quantile regression, with quantile random forests [1] as the underlying quantile regression algorithm.

Figure 2: Prediction intervals on simulated data constructed by locally adaptive conformal prediction, with conditional *median* estimation via quantile regression forests. The target coverage is 90%. On test data, the average coverage is 90.14% and the average length is 2.86.

To improve robustness to outliers, one might try to estimate the conditional median instead of the conditional mean in locally adaptive conformal prediction. We implement this strategy in Figure 2, using quantile regression forests [1] to estimate the conditional median. The residuals are scaled in the usual way, by classical regression via random forests. At least on this simulated dataset, estimating conditional medians instead of means has little effect on the average lengths of the prediction intervals (compare Figure 2 in this appendix with the subfigure for locally adaptive split conformal prediction in Section 4).

## A.2   Heavy-tailed Cauchy distribution

A second synthetic example highlights the advantage of CQR in handling data with outliers. We independently sampled $n = 2000$ univariate feature variables $X_i$ from the uniform distribution on the interval $[0,\ 10]$. We then sampled the response variable as:

$$Y_i \sim \text{Cauchy}(0, 6\sin^2(X_i)),$$

where $\text{Cauchy}(0, \gamma)$ is the Cauchy distribution with location parameter 0 and scale parameter $\gamma$. We also generated 5000 independent test samples from the same distribution, shown in Figure 3.

Figure 3: Cauchy distribution with varying scale parameter. Full range display of the test data.

As in the first synthetic experiment, we construct prediction intervals on test samples with target coverage of 90%. Figure 4 shows the prediction intervals obtained by the methods of (a) split conformal prediction, (b) locally weighted conformal with mean regression, (c) locally weighted conformal with median regression, and (d) CQR. In all cases, we divide the training data into two disjoint subsets, a proper training set and a calibration set. We fit a random forests regressor on the former and calibrate the intervals using the latter. To scale the residuals in the locally weighted version of split conformal inference, we fit another random forests regressor (conditional mean). In CQR, we use quantile random forests [1] to estimate the lower and upper conditional quantiles. The same regressor is used to estimate the conditional median in Figure 4c.

(a) Split: Avg. coverage 90.9%; Avg. length 98.94.

(b) Local: Avg. coverage 90.7%; Avg. length 77.64.

(c) Local: Avg. coverage 90.4%; Avg. length 59.78.

(d) CQR: Avg. coverage 90.2%; Avg. length 36.71.

Figure 4: Prediction intervals (target coverage rate is 90%) on simulated data, sampled from the Cauchy distribution: (a) standard split conformal method, (b) locally weighted split conformal (conditional mean), (c) locally weighted split conformal (conditional median), and (d) CQR. The broken black curves in (a) and (b) are the pointwise predictions from the random forests. In (c), the broken black curve is the conditional median estimate from the quantile random forests [1]. The two black curves in (d) are the lower and upper quantile regression estimates based on random forests.

In this experiment, the intervals constructed by the local conformal method (Figures 4b, 4c) are more efficient than those of split conformal (Figure 4a). Moreover, replacing the conditional mean regression (Figure 4b) with the conditional median (Figure 4c) improves the statistical efficiency of the local approach. However, CQR (Figure 4d) performs best on average, constructing the shortest intervals. All four methods yield intervals that attain the desired coverage rate of 90%, as expected.

## B  Lemmas about quantiles

Recall that the *quantile function* $Q$ of a random variable $Z$, with cumulative distribution function $F(z) := \mathbb{P}\{Z \le z\}$, is defined by the equivalence

$$Q(\alpha) \le z \quad \text{if and only if} \quad \alpha \le F(z)$$

for all $\alpha \in (0,1)$ and $z \in \mathbb{R}$. Dually, but less standardly, the *right quantile function* $R$ of the random variable $Z$ is defined by the equivalence

$$F^-(z) \le \alpha \quad \text{if and only if} \quad z \le R(\alpha),$$

where $F^-(z) := F(z-) = \mathbb{P}\{Z < z\}$. The quantile functions have the explicit formulas

$$Q(\alpha) = \inf\{z \in \mathbb{R} : \alpha \le F(z)\}, \qquad R(\alpha) = \sup\{z \in \mathbb{R} : F^-(z) \le \alpha\}.$$

As a special case, the *empirical quantile function* $\hat{Q}_n$ of random variables $Z_1, \dots, Z_n$ is the quantile function with respect to the empirical CDF $\hat{F}_n(z) := \frac{1}{n} \sum_{i=1}^n 1_{Z_i \le z}$. Likewise, the *right empirical quantile function* $\hat{R}_n$ of $Z_1, \dots, Z_n$ is the right quantile function with respect to $\hat{F}_n^-(z) = \frac{1}{n} \sum_{i=1}^n 1_{Z_i < z}$. They have the explicit formulas

$$\hat{Q}_n(\alpha) = Z_{(\lceil \alpha n \rceil)}, \qquad \hat{R}_n(\alpha) = Z_{(\lfloor \alpha n \rfloor + 1)},$$

where $Z_{(k)}$ denotes the $k$th smallest value in $Z_1, \ldots, Z_n$.

Variants of the following lemmas appear in the literature [2–4]. In the interest of clarity and a self-contained exposition, we state and prove them here.

**Lemma 1** (Quantiles and exchangeability). *Suppose $Z_1, \ldots, Z_n$ are exchangeable random variables. For any $\alpha \in (0, 1)$,*

$$\mathbb{P}\{Z_n \leq \hat{Q}_n(\alpha)\} \geq \alpha.$$

*Moreover, if the random variables $Z_1, \ldots, Z_n$ are almost surely distinct, then also*

$$\mathbb{P}\{Z_n \leq \hat{Q}_n(\alpha)\} \leq \alpha + \frac{1}{n}.$$

*In this statement, the probabilities are taken over all the variables $Z_1, \ldots, Z_n$.*

*Proof.* By exchangeability and the symmetry of $\hat{Q}_n(\alpha)$ as a function of $Z_1, \ldots, Z_n$, the probability $\mathbb{P}\{Z_i \leq \hat{Q}_n(\alpha)\}$ is equal to $\mathbb{P}\{Z_n \leq \hat{Q}_n(\alpha)\}$ for every $i$. Therefore,

$$\mathbb{E}\,\hat{F}_n(\hat{Q}_n(\alpha)) = \frac{1}{n}\sum_{i=1}^n \mathbb{P}\{Z_i \leq \hat{Q}_n(\alpha)\} = \mathbb{P}\{Z_n \leq \hat{Q}_n(\alpha)\}.$$

By the defining property of the quantile functions, $\hat{F}_n(\hat{Q}_n(\alpha)) \geq \alpha$ and $\hat{F}_n^-(\hat{R}_n(\alpha)) \leq \alpha$. Moreover, if the samples $Z_1, \ldots, Z_n$ are distinct, then $\|\hat{F}_n - \hat{F}_n^-\|_\infty \leq \frac{1}{n}$, and since $\hat{Q}_n \leq \hat{R}_n$, we have $\hat{F}_n(\hat{Q}_n(\alpha)) \leq \hat{F}_n(\hat{R}_n(\alpha)) \leq \hat{F}_n^-(\hat{R}_n(\alpha)) + \frac{1}{n} \leq \alpha + \frac{1}{n}$. To complete the proof, take expectations of the inequalities $\hat{F}_n(\hat{Q}_n(\alpha)) \geq \alpha$ and $\hat{F}_n(\hat{Q}_n(\alpha)) \leq \alpha + \frac{1}{n}$. $\qquad \square$

**Lemma 2** (Inflation of quantiles). *Suppose $Z_1, \ldots, Z_{n+1}$ are exchangeable random variables. For any $\alpha \in (0, 1)$,*

$$\mathbb{P}\{Z_{n+1} \leq \hat{Q}_n((1 + \tfrac{1}{n})\alpha)\} \geq \alpha.$$

*Moreover, if the random variables $Z_1, \ldots, Z_{n+1}$ are almost surely distinct, then also*

$$\mathbb{P}\{Z_{n+1} \leq \hat{Q}_n((1 + \tfrac{1}{n})\alpha)\} \leq \alpha + \frac{1}{n+1}.$$

*Proof.* Let $Z_{(k,m)}$ denote the $k$th smallest value in $Z_1, \ldots, Z_m$. Then for any $0 \leq k \leq n$, we have

$$Z_{n+1} \leq Z_{(k,n)} \quad \text{if and only if} \quad Z_{n+1} \leq Z_{(k,n+1)}.$$

Indeed, if $Z_{n+1} \leq Z_{(k,n)}$, then $Z_{(k,n+1)}$ is the larger of $Z_{(k-1,n)}$ and $Z_{n+1}$; in particular, $Z_{(k,n+1)} \geq Z_{n+1}$. Conversely, if $Z_{n+1} \leq Z_{(k,n+1)}$ then also $Z_{n+1} \leq Z_{(k,n)}$ because $Z_{(k,n+1)} \leq Z_{(k,n)}$.

Thus, since $\hat{Q}_n((1 + \tfrac{1}{n})\alpha) = Z_{(\lceil \alpha(n+1) \rceil, n)}$ and $\hat{Q}_{n+1}(\alpha) = Z_{(\lceil \alpha(n+1) \rceil, n+1)}$, we have

$$Z_{n+1} \leq \hat{Q}_n((1 + \tfrac{1}{n})\alpha) \quad \text{if and only if} \quad Z_{n+1} \leq \hat{Q}_{n+1}(\alpha)$$

and, hence,

$$\mathbb{P}\{Z_{n+1} \leq \hat{Q}_n((1 + \tfrac{1}{n})\alpha)\} = \mathbb{P}\{Z_{n+1} \leq \hat{Q}_{n+1}(\alpha)\}.$$

To conclude the proof, apply Lemma 1 with $n$ replaced by $n + 1$. $\qquad \square$

## C   Proofs of the main theorems

In this appendix, we prove the validity of the CQR prediction intervals described in Section 4.

**Theorem 1.** *If $(X_i, Y_i)$, $i = 1, \ldots, n+1$ are exchangeable, then the prediction interval $C(X_{n+1})$ constructed by the split CQR algorithm satisfies*

$$\mathbb{P}\{Y_{n+1} \in C(X_{n+1})\} \geq 1 - \alpha.$$

*Moreover, if the conformity scores $E_i$ are almost surely distinct, then the prediction interval is nearly perfectly calibrated:*

$$\mathbb{P}\{Y_{n+1} \in C(X_{n+1})\} \leq 1 - \alpha + 1/(|\mathcal{I}_2| + 1).$$

*Proof.* The result even holds, and we will prove it, conditionally on the proper training set.

Let $E_{n+1}$ be the conformity score

$$E_i := \max\{\hat{q}_{\alpha_{lo}}(X_i) - Y_i, Y_i - \hat{q}_{\alpha_{hi}}(X_i)\}$$

at the test point $(X_{n+1}, Y_{n+1})$. By the construction of the prediction interval, we have

$$Y_{n+1} \in C(X_{n+1}) \quad \text{if and only if} \quad E_{n+1} \leq Q_{1-\alpha}(E, \mathcal{I}_2),$$

and, in particular,

$$\mathbb{P}\{Y_{n+1} \in C(X_{n+1}) \,|\, (X_i, Y_i) : i \in \mathcal{I}_1\} = \mathbb{P}\{E_{n+1} \leq Q_{1-\alpha}(E, \mathcal{I}_2) \,|\, (X_i, Y_i) : i \in \mathcal{I}_1\}. \quad (2)$$

Since the original pairs $(X_i, Y_i)$ are exchangeable, so are the calibration variables $E_i$ for $i \in \mathcal{I}_2$ and $i = n+1$. Therefore, by Lemma 2 on inflated empirical quantiles (stated in Appendix B),

$$\mathbb{P}\{E_{n+1} \leq Q_{1-\alpha}(E, \mathcal{I}_2) \,|\, (X_i, Y_i) : i \in \mathcal{I}_1\} \geq 1 - \alpha, \quad (3)$$

and, under the additional assumption that the $E_i$'s are almost surely distinct,

$$\mathbb{P}\{E_{n+1} \leq Q_{1-\alpha}(E, \mathcal{I}_2) \,|\, (X_i, Y_i) : i \in \mathcal{I}_1\} \leq 1 - \alpha + \frac{1}{|\mathcal{I}_2| + 1}. \quad (4)$$

The result follows by taking expectations over the proper training set in (2), (3), and (4). □

Next, we prove the validity of the CQR prediction intervals that control the left and right tails independently.

**Theorem 2.** *Define the prediction interval*

$$C(X_{n+1}) := [\hat{q}_{\alpha_{lo}}(X_{n+1}) - Q_{1-\alpha_{lo}}(E_{lo}, \mathcal{I}_2), \; \hat{q}_{\alpha_{hi}}(X_{n+1}) + Q_{1-\alpha_{hi}}(E_{hi}, \mathcal{I}_2)],$$

*where $Q_{1-\alpha_{lo}}(E_{lo}, \mathcal{I}_2)$ is the $(1 - \alpha_{lo})$-th empirical quantile of $\{\hat{q}_{\alpha_{lo}}(X_i) - Y_i : i \in \mathcal{I}_2\}$ and $Q_{1-\alpha_{hi}}(E_{hi}, \mathcal{I}_2)$ is the $(1 - \alpha_{hi})$-th empirical quantile of $\{Y_i - \hat{q}_{\alpha_{hi}}(X_i) : i \in \mathcal{I}_2\}$. If the samples $(X_i, Y_i)$, $i = 1, \ldots, n+1$ are exchangeable, then*

$$\mathbb{P}\{Y_{n+1} \geq \hat{q}_{\alpha_{lo}}(X_{n+1}) - Q_{1-\alpha_{lo}}(E_{lo}, \mathcal{I}_2)\} \geq 1 - \alpha_{lo} \quad (5)$$

*and*

$$\mathbb{P}\{Y_{n+1} \leq \hat{q}_{\alpha_{hi}}(X_{n+1}) + Q_{1-\alpha_{hi}}(E_{hi}, \mathcal{I}_2)\} \geq 1 - \alpha_{hi}. \quad (6)$$

*Consequently, assuming $\alpha = \alpha_{lo} + \alpha_{hi}$, we also have $\mathbb{P}\{Y_{n+1} \in C(X_{n+1})\} \geq 1 - \alpha$.*

*Proof.* The two events inside the probabilities (5) and (6) are equivalent to $\hat{q}_{\alpha_{lo}}(X_{n+1}) - Y_{n+1} \leq Q_{1-\alpha_{lo}}(E_{lo}, \mathcal{I}_2)$ and $Y_{n+1} - \hat{q}_{\alpha_{hi}}(X_{n+1}) \leq Q_{1-\alpha_{hi}}(E_{hi}, \mathcal{I}_2)$, respectively. We can thus apply Lemma 2 twice, in the same manner as in the proof of Theorem 1. □

# D  Experiments

In this appendix, we describe in greater detail the methods and datasets employed in the experiments of Section 6. The source code implementing the experiments is available for download externally.[1] We used ClusterJob [5] to manage and run the experiments on our local cluster.

## D.1  Methods

In our experiments, we compare the following methods related to conformal prediction. First, we evaluate the original version of split conformal prediction (Section 3) using the following three regression algorithms.

- **Ridge**: We include ridge regression as a baseline. The regularization parameter is tuned by cross validation.
- **Random Forests**: We use the implementation of (conditional mean) random forest regression in the Python package `sklearn`. The hyper-parameters are the package defaults, except for the total number of trees in the forest, which we set to 1000.

- **Neural Net**: Our neural network architecture consists of three fully connected layers, with ReLU nonlinearities between layers. The first layer takes as input the $p$-dimensional feature vector $X$ and outputs $64$ hidden variables. The second layer follows the same template, outputting another $64$ hidden variables. Finally, a linear output layer returns a pointwise estimate of the response variable $Y$. The parameters of the network are fit by minimizing the mean squared error loss function. We use the stochastic optimization algorithm Adam [6], with fixed learning rate of $5 \times 10^{-4}$, minibatches of size $64$, and weight decay parameter equal to $10^{-6}$. We employ dropout regularization [7], with the probability of retaining a hidden unit equal to $0.1$. To avoid overfitting, we found that early stopping performs well; we tune the number of epochs by cross validation, with an upper limit of $1000$ epochs.

We evaluate locally adaptive conformal prediction (Section 5) using the same three underlying regression algorithms. Practitioners employ various tweaks to improve the method's numerical stability and statistical performance. Following [8], we add a hyper-parameter $\gamma > 0$ as a constant offset to the scale estimator $\hat{\sigma}(x)$. The scaled residuals then become $\tilde{R}_i = R_i / (\hat{\sigma}(X_i) + \gamma)$. We set the hyper-parameter $\gamma$ to $1$, which improves performance considerably compared to $\gamma = 0$.

- **Ridge Local**: The conditional mean estimator $\hat{\mu}$ is fit by ridge regression, as described above, and the mean absolute deviation (MAD) estimator $\hat{\sigma}$ is $k$-nearest neighbors with $k = 11$.
- **Random Forests Local**: Both $\hat{\mu}$ and $\hat{\sigma}$ are random forests with the hyper-parameters described above.
- **Neural Net Local**: Both $\hat{\mu}$ and $\hat{\sigma}$ are neural networks, with the network architecture, hyper-parameters, and training algorithm described above.

For our own proposal, conformalized quantile regression (Section 4, Algorithm 1), we evaluate two variants:

- **CQR Random Forests**: We use CQR with quantile regression forests [1]. To ensure a fair comparison, the hyper-parameters of the quantile regression forests are made identical to those of the random forests in the previous methods. Quantile regression forests have two additional parameters that control the coverage rate on the training data. We tune them using cross validation, as explained in Section 4.
- **CQR Neural Net**: We apply CQR using neural networks for quantile regression [9]. The network architecture is the same as above, except that the output of the quantile regression network is a two-dimensional vector, representing the lower and upper conditional quantiles. The training algorithm is also the same, except that the cost function is now the pinball loss instead of the quadratic loss.

Finally, for the sake of comparison, we also include the previous two quantile regression algorithms, but without any conformalization:

- **Quantile Random Forests**: We use quantile regression forests with hyperparameters as in the CQR procedure, except that the upper and lower levels are fixed at $0.05$ and $0.95$.
- **Quantile Neural Net**: We use quantile regression neural networks with exactly the same architecture and training algorithm as in the CQR procedure.

Unlike the preceding methods, the last two methods do not need a calibration set and do not have a finite-sample coverage guarantee. We fit them on the entire training set.

### D.2  Performance on individual datasets

In a series of figures, we break down the performance of the different methods on each of the benchmark datasets. Figure 5 summarizes our experiments on the datasets: medical expenditure panel survey number 19 (`MEPS_19`) [10], number 20 (`MEPS_20`) [11], and number 21 (`MEPS_21`) [12]. Figure 6 shows the results for: blog feedback (`blog_data`) [13]; physicochemical properties of protein tertiary structure (`bio`) [14]; and bike sharing (`bike`) [15]. Figure 7 shows the results for: community and crimes (`community`) [16]; Tennessee's student teacher achievement ratio (`STAR`) [17];

|  | MEPS_19 [10] | MEPS_20 [11] | MEPS_21 [12] | community [16] | STAR [17] |
|---|---|---|---|---|---|
| **# features** | 139 | 139 | 139 | 100 | 39 |
| **# samples** | 15785 | 17541 | 15656 | 1994 | 2161 |

|  | facebook_1 [19] | facebook_2 [19] | concrete [18] | blog_data [13] | bio [14] | bike [15] |
|---|---|---|---|---|---|---|
| **# features** | 53 | 53 | 8 | 280 | 9 | 18 |
| **# samples** | 40948 | 81311 | 1030 | 52397 | 45730 | 10886 |

Table 1: Dimensions of the benchmark datasets

and concrete compressive strength (`concrete`) [18]. Lastly, Figure 8 shows the results for: Facebook comment volume, variants one (`facebook_1`) and two (`facebook_2`) [19, 20].

The CQR random forests are overly conservative on the two Facebook datasets. This is consistent with the theory, because in this case there are ties among the conformity scores and so the upper bound in Theorem 1 does not apply.

### D.3 Further information about the datasets

The Medical Expenditure Panel Survey (MPES) data is subject to copyright and usage rules. The three datasets `MEPS_19`, `MEPS_20`, `MEPS_21` were downloaded from [10–12]. The feature and response variables were extracted using IBM's AIF 360 software package, available online [21].

We applied routine data cleaning and preprocessing to the other datasets. This is recorded in complete detail in the code that we have provided. For convenience, the dimensions of the benchmark datasets are listed in Table 1.

## Footnotes

[1] `https://github.com/yromano/cqr`

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

Figure 5: Average length (left) and coverage (right) of prediction intervals ($\alpha = 0.1$), averaged over 20 random (80%/20%) training/test splits. The numbers in the colored boxes are the average lengths, shown in red for split conformal, in gray for locally adapative split conformal, and in light blue for our method. The name of the dataset is located at the top of each plot.

Figure 6: Refer to the caption of Figure 5 for details.

Figure 7: Refer to the caption of Figure 5 for details.

Figure 8: Refer to the caption of Figure 5 for details.