[Reviews · NeurIPS 2019]

Reviewer 1



The paper is well written and the topic is highly relevant is this era of reproducibility crisis. However, its significance remains unclear wrt to the actual literature. 1- The classical conformal prediction method (section 3) is already valid for *any* regression algorithm (including quantile regression) and *any* conformity score function (including E_i in Eq. 6) (under mild assumptions e.g. exchangeability of the data and permutation invariance of the score). In that sense, the novelties of this paper rather limited and seems to be a direct application of classical conformal prediction with a quantile regression algorithm. If it is not the case, it should be interesting that the authors precisely clarify the fundamental differences between these approaches (More, the proofs seems to be exactly the same). Relatedly, the sentence in line 61-63 seems confusing. As far as I understand, standard method is not only restricted to conditional mean regression. Same remark for line 58-60, since any estimator can be used, the coverage guarantee hold regardless of the choice or accuracy (which one: accuracy in optimization or in prediction?) of the regression algorithm (also under mild assumptions). 2- To me, the discussions on the limitation in length of C(X_n+1) in Eq. 5 is also confusing. a) When one consider *full* conformal prediction set (not necessarily an interval) [mu(X_n+1) - Q_{1-alpha}(R), mu(X_n+1) + Q_{1-alpha}(R)] for all possibilities of y_n+1, the quantiles Q_{1-alpha}(R) depends on X_n+1 and its length is *not* fixed (in cases where it is an interval). Hence this limitation seems to come from the splitting strategy itself. b) The proposed conformal set C(X_n+1) in Eq. 7, has a length that is lower bounded by 2 Q_{1-alpha}(E, I_2) independently of X_n+1. Why the critics above does not hold in this case? Perhaps the locally adaptive approach can alleviate the issue. If the above points hold, the numerical experiments rather highlight the critical importance of the choice of the estimator and the conformity score used, to obtain small confidence set.

Reviewer 2



The paper is novel, well-written and important. I do not have any complaint but a few minor suggestions. (1) There are some works on linear quantile regression, which provided the non-conformalized version of the method in this paper, e.g. Zhou and Portnoy (1996, 1998). These may be worth mentioning. (2) The acronym CQR has been used for composite quantile regression (Zou and Yuan), which is another influential paper. It may be worth changing the acronym as this paper is likely to become influential too! (3) I like the point that conformalized quantile regression is better than locally adaptive conformal prediction. It may be better to design numerical experiments with highly heavy-tailed distribution (say Cauchy) to further illustrate this point. References Zhou, K. Q., & Portnoy, S. L. (1996). Direct use of regression quantiles to construct confidence sets in linear models. The Annals of Statistics, 24(1), 287-306. Zhou, K. Q., & Portnoy, S. L. (1998). Statistical inference on heteroscedastic models based on regression quantiles. Journal of Nonparametric Statistics, 9(3), 239-260. Zou, H., & Yuan, M. (2008). Composite quantile regression and the oracle model selection theory. The Annals of Statistics, 36(3), 1108-1126.

Reviewer 3



The paper provides an interesting link between Quantile Regression and Conformal Prediction that would allow to combine the strengths of both when dealing with heteroscedastic data. Both methods provide different ways of constructing prediction intervals. Quantile Regression portion of the method can use any QR method including Linear Quantile Regression, Quantile Random Forests, etc depending on the problem at hand. The experiment section demonstrated the tighter prediction intervals obtained by CQR Random Forests and CQR Neural Net.

[Author Response · NeurIPS 2019]

We thank the reviewers and the editor for their reviews and helpful comments, which will improve our manuscript. We are gratified to see Reviewer 2 (R2) write that "this paper is likely to become influential" and "is novel, well-written and important." We take seriously R2's suggestion to change our method's acronym and thank him/her for pointing out several references, which we have added to the paper. We have also followed the reviewer's suggestion to illustrate CQR using the heavy-tailed Cauchy distribution. As predicted by R2, this experiment indeed shows a clear advantage of CQR over split and locally adaptive conformal prediction. We have added this example to our manuscript.

Reviewer 1 (R1) observes that, under mild conditions, classical conformal prediction is valid for any regression algorithm and any conformity score function. This is obviously true and is noted in Section 5 of the manuscript. Therefore, the crucial question is this: **which regression algorithm and conformity score should one then use?** In this respect, our work marks a significant departure from the whole body of research built on the original version of conformal prediction, for *conditional mean regression*. We argue that the new types of conformity scores we develop improve significantly on the state of the art. We cannot put it better than R2: "although it [the use of quantile regression] appears to be a simple modification from hindsight, it is non-trivial from foresight because quantile based methods is highly adaptive to the heteroscedasticity, or more generally distributional heterogeneity, which is ubiquitous in real-world applications." This is the main point of our paper. Conformal inference is a beautiful idea but what if it had been implemented with subpar tools all along? In particular, why estimate the mean if the goal is to estimate quantiles?

To understand the limitations of classical conformal prediction, consider heteroscedastic data with outliers. Suppose we have complete knowledge of the conditional distribution $Y|X$, so that no learning is required. In this idealized setting, the usual conformity score becomes $R_i = |Y_i - \mu(X_i)|$, where $\mu$ is the *true* conditional mean regression function. This score would never yield optimal prediction intervals on heteroscedastic data! It is optimal only in the restrictive setting of a location model $Y = \mu(X) + \epsilon$, where the noise $\epsilon$ follows a symmetric, unimodal density function [1]. As for locally adaptive split conformal prediction, we argue in our paper that scaling residuals by their variance is also suboptimal, even if its limitations are less severe. In the present idealized setting, this method can only construct intervals that are adaptive to the location and local variance in $Y|X$. Distributional heterogeneity does not necessarily arise from such a location-scale family, as illustrated in our synthetic simulation (Figure 1). Furthermore, the locally adaptive score performs poorly on data with outliers, as confirmed by R2, who kindly guided us to design an additional experiment that corroborates this point.

In the ideal setting where the conditional distribution is known, *CQR would construct exact prediction intervals reflecting the intrinsic predictive uncertainty, while achieving any desired coverage level*. To see this, recall that the endpoints of the CQR prediction interval would be the true lower and upper conditional quantiles, and so the correction term $Q_{1-\alpha}(E, \mathcal{I}_2)$ would be equal to zero. This property stands in contrast to *all previous conformal prediction methods, which have generally nonzero correction terms even in the idealized setting*. (We thank R1 for raising concerns about the novelty of our paper and its importance; we have modified our manuscript to include this crucial discussion.) When the conditional distribution is unknown, our method improves accuracy. To quote R2: "[CQR] may influence the machine learning architecture for problems [with] continuous outcome. For instance, the last layer is typically the L2 loss, which gives no easy way to assess decision uncertainty with theoretical guarantee. However, replacing the last layer by pinball loss with two different quantiles would automatically provide an assessment of uncertainty and the conformalization step provides the theoretical guarantee in finite samples even in presence of arbitrary model misspecification."

R1 comments that the length of the prediction intervals constructed by the full conformal method is not fixed, whereas in split conformal it is equal to $2Q_{1-\alpha}(R, \mathcal{I}_2)$. This is true. To quote from [1], however: "for full conformal, the width can vary slightly as $X$ varies, but the difference is often negligible as long as the fitting method is moderately stable." We added a similar comment to our paper. That being said, the enormous computational cost of full conformal compared to split conformal means that *any comparison between the two is of very limited practical interest*.

R1 also wonders why the criticism about the fixed-length interval does not apply to CQR, as its intervals have length *lower bounded* by $2Q_{1-\alpha}(E, \mathcal{I}_2)$, independently of $X_{n+1}$. Our Figure 1 clearly shows that the intervals constructed by CQR do *not* have fixed length. As to why, notice that the interval length for split conformal is *equal to*, not merely lower bounded by, $2Q_{1-\alpha}(R, \mathcal{I}_2)$, whereas in our case, the quantity $Q_{1-\alpha}(E, \mathcal{I}_2)$ can be positive, zero, or even negative, depending on the calibration of the quantile regression method. When perfectly calibrated, as in the ideal setting, this quantity will be zero, making the lower bound trivial. We appreciate the reviewer's suggestion to integrate the locally adaptive approach into our framework. However, we are doubtful that doing so would improve its performance, since, as R2 notes, our paper "provides extensive high-quality numerical experiments, which clearly demonstrate the superior performance and adaptivity of conformalized quantile regression."

Reviewer 3 suggests that we construct a loss function formulating CQR as an optimization problem. In fact, we use the pinball loss function to estimate the conditional quantiles, which can be viewed as CQR's objective function.

[1] Lei et al. Distribution-free predictive inference for regression. *JASA*, 113(523):1094–1111, 2018.


[Meta-Review · NeurIPS 2019]

Reviewers are mostly favorable. Quantile regression is used widely and successfully in practice, so giving it stronger theoretical guarantees is worthwhile. The key issue in the less favorable review is in the quote from [1] by Vovk et al., 1999: "The full conformal and split conformal methods both tend to produce prediction bands C(x) whose width is roughly constant over x in Rd. ... in some scenarios ... the residual variance will vary nontrivially with X, and in such a case we want the conformal band to adapt correspondingly." Heteroskedasticity is an important issue that for 20 years hasn't been addressed fully in the context of conformal prediction. As the reviewer says, "mu(X) can be any regression estimator including the quantile regression proposed in this paper." The contribution of this submission is to work out this idea in detail, which is a good contribution.